# Boosted Long Short-Term Memory with Additional Inner Layers

## Abstract

Long Short-Term Memory (LSTM) is widely known as a powerful type of Recurrent Neural Network, allowing it to achieve great results on many difficult sequential data tasks. Numerous experiments have shown that adding more complexity to neural network architectures may lead to a significant increase in performance that outweighs the incurred costs of an upgraded structure. In this paper, we propose a Boosted LSTM model created by adding layers inside the LSTM unit to optimize the model by enhancing its memory and reasoning capabilities. We evaluated the performance of different versions of Boosted LSTM architectures using three empirical tasks, studying the impact of different placements of additional layers, the activation functions used in the additional layers, and the model's hidden units. The experiments have shown that the Boosted LSTM unit, which uses Exponential Linear Unit as its boosted layers activation function, performs better than the similar models created from the simple LSTM units while often taking fewer epochs to achieve similar or better results, usually in a smaller number of training epochs.

## 1 Introduction

Recurrent Neural Networks (RNNs) are devoted to processing sequential data. Since input order is crucial for this kind of data, networks that tackle this problem must have mechanisms to memorize sequential relationships through possibly many periods of time. Long Short-Term Memory (LSTM) was created specifically to allow for learning long-term dependencies by eliminating the vanishing gradient problem, which hindered the simple RNNs' ability to work efficiently (Bengio et al., 1994; Hochreiter, 1991; Hochreiter & Schmidhuber, 1997).

LSTMs have a more complex architecture, with a constant error carousel, internal memory, and multiplicative gates, which all enhance unit capacity to store information efficiently. To this day, they find numerous applications in tasks such as handwriting (Lopez-Rodriguez et al., 2022), (Misgar et al., 2022) and speech recognition (Abdelhamid et al., 2022), text analysis tasks (Zhao et al., 2022), price forecasting (Bukhari et al., 2020), (Zha et al., 2022) and even epidemic dynamic modeling (Shahid et al., 2020).

When considering architecture-related LSTM modifications, there are two apparent approaches. The first is to trim, rewire, and reduce it in the hope of achieving better results, or at least not degrading the network performance while reducing the training time (Greff et al., 2017), (Lu, 2016). One of the most prominent results of this approach is the Gated Recurrent Unit (GRU) (Cho et al., 2014). The second is to add and stack more depth to the existing architecture, a way which, among others, resulted in the appearance of Bidirectional LSTMs (Graves & Schmidhuber, 2005), Grid LSTMs (Kalchbrenner et al., 2015), and Nested LSTMs (Moniz & Krueger, 2017).

We took the second path by adding more layers and more depth and creating a new LSTM architecture called Boosted LSTM. This structure enhanced by additional layers allows for more memory depth and sophisticated reasoning when processing information. We tested different versions of the proposed model on three empirical tasks to discover the one with the greatest potential to improve efficiency. Our experiments demonstrated that the final Boosted LSTM architecture not only performs better but also often achieves better results in fewer epochs than the classic LSTM.

## 2 METHODS

Since 1997, when LSTM was introduced by Hochreiter & Schmidhuber (1997), its architecture has undergone numerous changes, such as the addition of a forget gate so that the network state could be reset (Gers et al., 2000), the addition of peephole connections to allow LSTM gates to access the Cell State (Gers et al., 2003), nesting LSTM units to add multiple levels of memory (Moniz & Krueger, 2017) and many others.

### 2.1 ARCHITECTURE

As a base for further experiments, we used the LSTM architecture that was often used in the literature (Greff et al., 2017), (Lu, 2016), (Moniz & Krueger, 2017), with units similar to those defined by Graves (2013). We will refer to it as the Vanilla LSTM cell (Fig. 1a), defined by the following internal operations:

$$i_t = \sigma(W_{ih}h_{t-1} + W_i x_t + b_i) \tag{1}$$

$$f_t = \sigma(W_{fh}h_{t-1} + W_f x_t + b_f) \tag{2}$$

$$o_t = \sigma(W_{oh}h_{t-1} + W_o x_t + b_o) \tag{3}$$

$$\bar{c}_t = tanh(W_{ch}h_{t-1} + W_c x_t + b_c) \tag{4}$$

$$c_t = f_t \otimes c_{t-1} + i_t \otimes \bar{c}_t \tag{5}$$

$$h_t = o_t \otimes tanh(c_t) \tag{6}$$

where $i_t$, $f_t$, and $o_t$ are the input, forget, and output gate activation vectors respectively, $\bar{c}_t$ is the candidate vector, $c_t$ is the Cell State vector, $h_t$ is the hidden unit signal vector, and $t$ is a time step. $W$ represents the weight matrix, where the subscript denotes its association, e.g., $W_{ih}$ being the weight matrix for the incoming hidden unit in the input gate equation. A similar rule applies for biases, which are denoted as $b_i$, $b_f$, $b_o$, and $b_c$. Operator $\otimes$ denotes the Hadamard (element-wise) multiplication of the vectors.

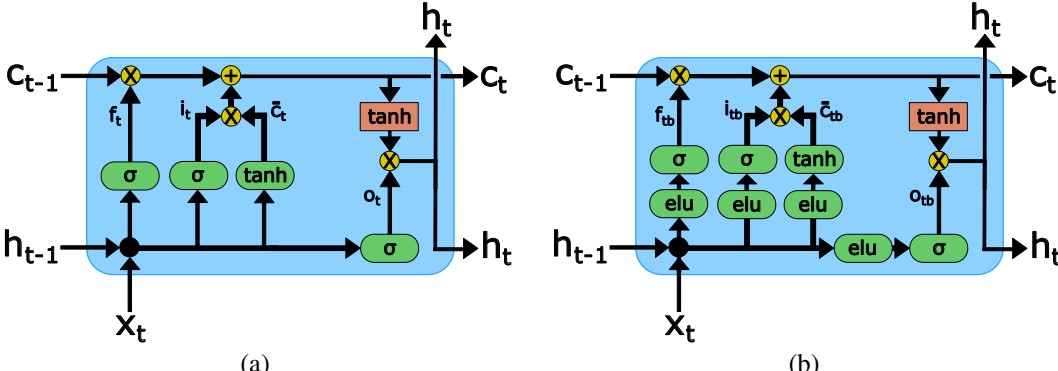

(a)            (b)

Figure 1: Comparison of (a) Vanilla LSTM and (b) Boosted LSTM Units using Exponential Linear Unit (ELU) as the activation function for the additional boosting layers.

Boosting of the LSTM unit (Fig. 1b) was performed by introducing additional layers to its architecture. Fig. 1 illustrates a comparison of classic and boosted LSTM memory units, where the additional boosting layers are placed just before the base ones, as presented in the following formulas:

$$i_{tb} = \sigma(W_{ib}(elu(W_{ih}h_{t-1} + W_i x_t + b_i)) + b_{ib}) \tag{7}$$

$$f_{tb} = \sigma(W_{fb}(elu(W_{fh}h_{t-1} + W_f x_t + b_f)) + b_{fb}) \tag{8}$$

$$o_{tb} = \sigma(W_{ob}(elu(W_{oh}h_{t-1} + W_o x_t + b_o)) + b_{ob}) \tag{9}$$

$$\bar{c}_t = tanh(W_{cb}(elu(W_{ch}h_{t-1} + W_c x_t + b_c)) + b_{cb}) \tag{10}$$

$$c_{tb} = f_{tb} \otimes c_{t-1} + i_{tb} \otimes \bar{c}_t \tag{11}$$

$$h_t = o_{tb} \otimes tanh(c_{tb}) \tag{12}$$

The main change in equations (7), (8), (9), and (10) is the introduction of the boosting layers with their own weights ($W_{ib}, W_{fb}, W_{ob}, W_{cb}$) and biases ($b_{ib}, b_{fb}, b_{ob}, b_{cb}$). Based on this change, several different architectures, using $ELU$ (Exponential Linear Unit), $SELU$ (Scaled Exponential Linear Unit), and $ReLU$ (Rectified Linear Unit) before one or more base layers, as potential activation functions were considered.

Inevitably, such architecture modifications result in a change in the computational complexity of these boosted units in comparison to the classic LSTM unit. Considering the time complexity of Vanilla LSTM, we can calculate it by computing the time complexity of the input gate using equation 1, since it will be identical for the rest of the gates. Assuming that $k$ is the size of the input vector and $h$ is the size of the hidden layer, we may estimate the time complexities inside the equation:

- $W_{ih}h_{t-1}$ has a time complexity of $O(h^2)$,
- $W_i x_t$ has a time complexity of $O(kh)$,
- $W_i x_t + b_i$ has a time complexity of $O(h)$,
- applying activation function (sigmoid/tanh) has a time complexity of $O(h)$.

After taking all these factors into consideration, we obtained the computational complexity of the Vanilla LSTM gates:
$$O(4(h^2 + kh + 2h)) = O(4h(h + k + 2)) \tag{13}$$

Summing up, by considering the equation 5 and 6, each adding the $O(2h)$ complexity, we obtain the computational efficiency of the Vanilla LSTM cell equal to

$$O(4h(h + k + 3)) \tag{14}$$

For the Boosted LSTM, the internal part is identical as in the case of the Vanilla LSTM, but we need to add in the complexity of the added boosted layers:

- $W_{ib}$ multiplication with the inner layer output has a time complexity of $O(h^2)$,
- adding the biases to the result has a time complexity of $O(h)$,
- applying $ELU$ activation function has a time complexity of $O(h)$.

Considering all four gates and equations 11 and 12 the time complexity of the Boosted LSTM is

$$O(4h(2h + k + 5)) \tag{15}$$

Therefore, Boosted LSTM requires more resources to run, both memory and computational time-wise and its performance must justify the additional commitment.

## 2.2 DATASETS AND MODELS

The performance of the architecture was verified on three datasets: the IMDB sentiment classification dataset (Maas et al., 2011), the Permuted Sequential MNIST dataset (Lecun et al., 1998), and the Reuters Newswire classification dataset (Apté et al., 1994). This allowed us to test different capabilities of the architecture while maintaining focus on the text inference tasks. For all tested models, train and test subsets were obtained via the Keras API, while the validation subset was created by taking 20% of the training subset. Other hyperparameters were adapted accordingly to the dataset considered, taking the Vanilla LSTM performance as a reference point for their refinement.

IMDB (Internet Movie Database) introduced by Maas et al. (2011) dataset consists of 50,000 movie reviews, labeled according to the reviewer's sentiment (positive or negative). Both training and testing sets consist of 25,000 reviews encoded as sequences of word indexes. From the training set 20% of all examples were removed to create the validation set. We experimentally choose the top 4000 most common words with a maximum sequence length of 300, leading to zero-padding of shorter sequences and truncating of longer ones. The RMSprop optimizer with a learning rate of $1e^{-4}$ and binary cross-entropy loss was used. The model consisted of an embedding layer, an appropriate LSTM layer, a dense layer of 8 units with a ReLU activation function, and a final dense layer with 1 unit and a sigmoid activation function. Both after the embedding layer and after the LSTM layer, $0.25$ dropouts were used, and the l2 regularization with the factor $1e^{-4}$ was applied

to this LSTM layer. A batch size of 512 was chosen for the experiments carried out. Tests were performed for models with 2, 4, and 8 hidden units.

The MNIST (Modified National Institute of Standards and Technology) dataset by Lecun et al. (1998) contains 70,000 examples of handwritten digits, where 48,000 images were used for training, 12,000 for validation and 10,000 images for testing. Each image in the dataset is in grayscale of 28x28 pixel size. Additionally, a fixed random permutation was applied to each image pixel before feeding it to the model (hence we will refer to it as Permuted MNIST). This results in the dislocation of patterns, the removal of any local spatial correlations, and an overall increase in task difficulty. This shifts the focus more on testing the models' ability to handle long-term dependencies, making it more appropriate for RNN performance evaluation tasks. Based on Zhou et al. (2016), there are two approaches to the organization of the LSTM input for these data. The first is to flatten the image to a sequence of length 784; and the second treats each row of the image as an individual 28-pixel-long vector, yielding 28 input sequences. During the tests, the RMSprop optimizer with a learning rate of $1e^{-3}$ was used alongside the sparse categorical cross-entropy as a loss function. In the LSTM unit, the regularization l2 with the probability of $1e^{-4}$ and $0.25$ dropout was used. The final classification was performed by a simple dense layer of 10 units with a softmax activation function. A batch size of 512 was used during the training process. Hidden units of 32, 64, and 128 sizes were used during the experiments.

The Reuters Newswire dataset (Apté et al., 1994) consists of 11,228 newswires from the Reuters news agency, labeled over 46 different topics. The training dataset has 8,982 encoded newswires (as sequences of word indices), while 2,246 are used in the test dataset. We took 1,796 examples from the training set to create a validation set. In our experiments, we considered 2,000 words as features and set the maximum sequence length to 300. A simple model, consisting of the input embedding layer, the appropriate LSTM layer, the dense output layer of 46 units, and the softmax activation function, was complied with the RMSprop optimizer with the $1e^{-3}$ learning rate and the sparse categorical cross-entropy loss function. A dropout of $0.25$ was applied after the embedding and LSTM layers, where for the latter, the regularization of l2 of $1e^{-4}$ was also applied. A batch size of 512 and a variable number of hidden units (8, 16, 32) were assumed.

## 3 Experiments and Results

Initial experiments focused on checking the performance of five different architectures, specifically, Fully Boosted LSTM, LSTM with Boosted Input Gate, LSTM with Boosted Forget Gate, LSTM with Boosted Output Gate, and LSTM with Boosted Cell Gate. All results were compared with the reference models (Vanilla LSTM, GRU, and Bidirectional LSTM) and each other. After testing, the architectures where only one of the gates was boosted appeared to be the worst of the group. Just increasing the complexity of the gating mechanism on its own did not produce better results.

For the most promising architectures, the influence of the boosted activation function was evaluated, and in the end, the Boosted LSTM with ELU boosted activation function, the Boosted Cell State LSTM with ReLU boosted activation function, and the Boosted Cell State LSTM with ELU boosted activation function emerged as the best candidates for further experiments. The final precision of the models was evaluated with the dedicated test datasets, independent of both the train and the validation datasets.

### 3.1 IMDB Dataset

Assessment of the best candidates for the IMDB dataset was hindered by the appearance of overfitting during the 300-epoch training process. As seen in Fig. 3, this was problematic when the considered architectures were based on ELU as a boosted activation function, as they were constantly converging at a faster rate (30.15% on average) than the rest of the models. The most severe case of this was for 8 hidden unit-based models. After analyzing the training histories, we decided to apply early stopping criteria to preserve the best results for each architecture.

Table 1: The best accuracies and losses achieved for the final models of the IMDB sentiment classification task.

| LSTM Architecture | Accuracy | | | Loss | | |
|---|---|---|---|---|---|---|
| | 2 Units | 4 Units | 8 Units | 2 Units | 4 Units | 8 Units |
| Vanilla LSTM | 0.8713 | 0.8719 | 0.8646 | 0.3127 | 0.3322 | 0.3569 |
| GRU | 0.8730 | 0.8715 | 0.8754 | 0.3086 | 0.3177 | 0.3102 |
| Bidirectional LSTM | 0.8732 | 0.8690 | 0.8807 | 0.3097 | 0.3322 | 0.2955 |
| Boosted LSTM ELU | 0.8778 | 0.8798 | 0.8816 | 0.2982 | 0.3494 | 0.2984 |
| Boosted Cell State ReLU | 0.8665 | 0.8616 | 0.8798 | 0.3310 | 0.3539 | 0.2975 |
| Boosted Cell State ELU | 0.8704 | 0.8728 | 0.8714 | 0.3209 | 0.3209 | 0.3206 |

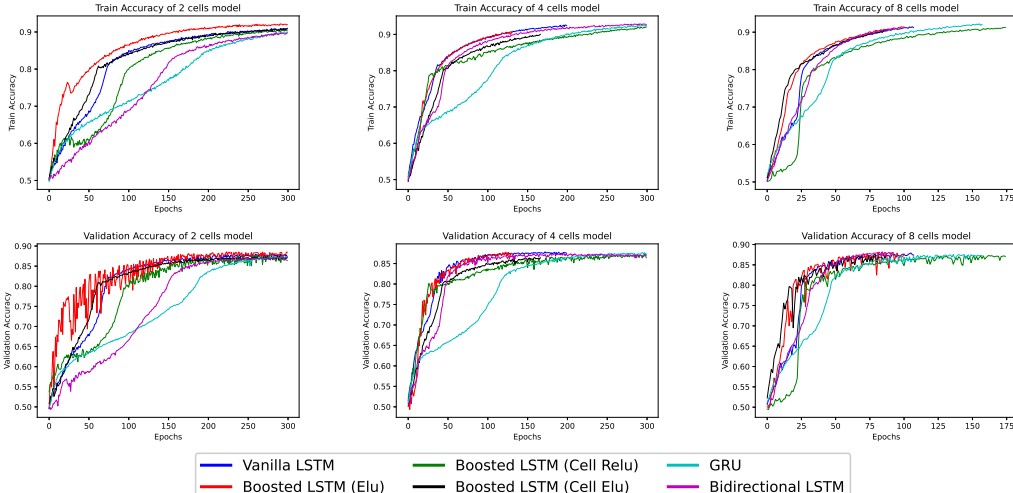

Figure 2: Accuracy vs. epochs for the train and validation datasets for all considered IMDB sentiment classification task architectures. The difference in epochs is due to the response to the implemented early stopping criterion.

The final results are summarized in Table 1. The fully boosted model based on the ELU-boosted activation function managed to outperform all the other candidates in terms of its accuracy on the test datasets. Its final loss for 2 and 8 units is also relatively small, being only worse than for Bidirectional LSTM and LSTM with Cell State boosted with ReLU boosted activation function.

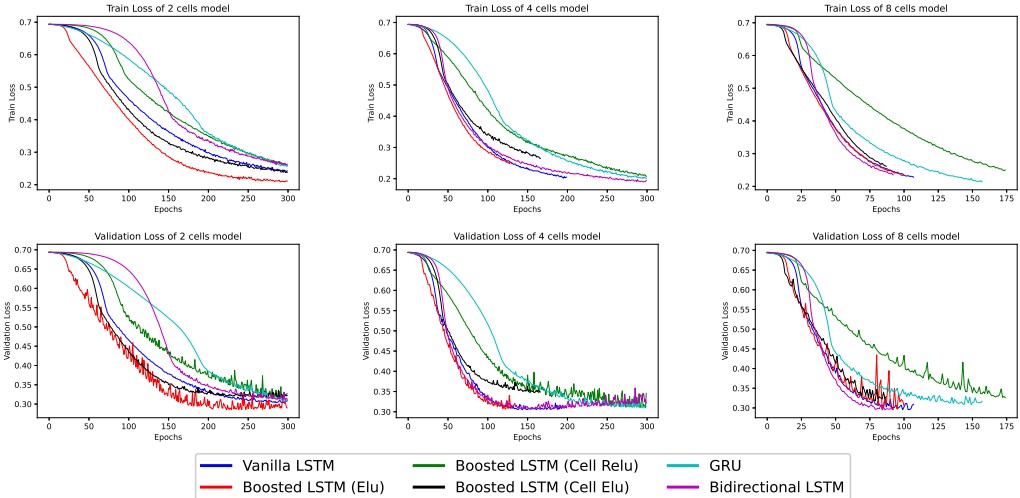

Figure 3: Loss vs. epochs for the train and validation datasets for all considered IMDB sentiment classification task architectures. The difference in epochs is due to the response to the implemented early stopping criterion.

## 3.2 PERMUTED MNIST DATASET

The row-wise training of candidate models on the MNIST dataset lasted for 500 epochs. Looking at the plots in Fig. 5, we found that the Boosted LSTM with the ELU-boosted activation function managed to constantly score the best validation results while maintaining one of the fastest convergence rates.

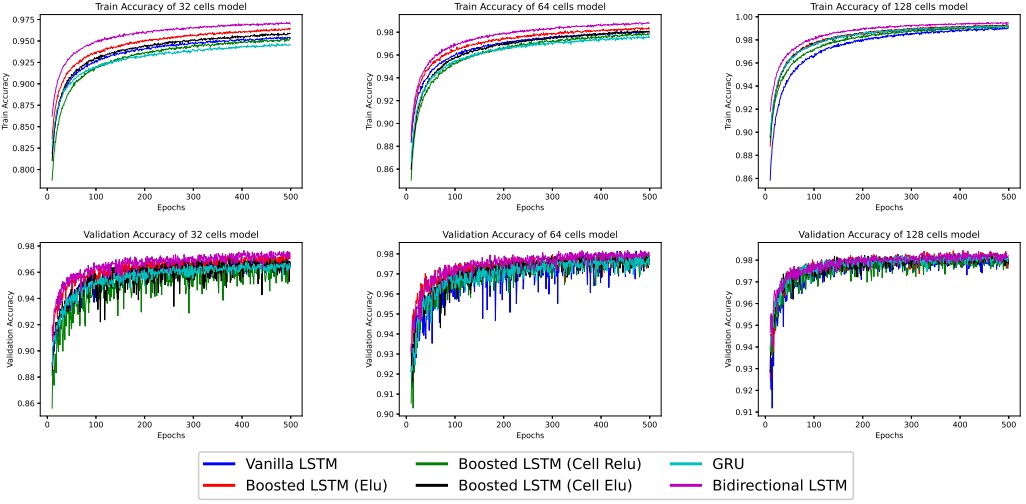

Figure 4: Accuracy vs. epochs for the train and validation datasets for all the considered permuted MNIST classification task architectures.

The improvement in accuracy is, seemingly, not by a large margin, as shown in Table 2, the greatest increase of 0.72% in relation to the Vanilla LSTM occurred for the 32-unit architecture, but since we are operating so close to 100%, it is still a valuable improvement. For all tests, Boosted LSTM with ELU-boosted activation function scored the best results, both accuracy and loss-wise, losing marginally with the GRU for the 128-unit architecture. Boosting the LSTM Cell State with neither ReLU nor ELU yields non-negligibly better results.

Table 2: Best accuracy and losses for the final models of the permuted MNIST classification task.

| LSTM Architecture | Accuracy | | | Loss | | |
|---|---|---|---|---|---|---|
| | 32 Units | 64 Units | 128 Units | 32 Units | 64 Units | 128 Units |
| Vanilla LSTM | 0.9639 | 0.9774 | 0.9803 | 0.1312 | 0.0858 | 0.0873 |
| GRU | 0.9625 | 0.9772 | 0.9811 | 0.1365 | 0.0861 | 0.0830 |
| Bidirectional LSTM | 0.9704 | 0.9789 | 0.9805 | 0.1065 | 0.0842 | 0.0953 |
| Boosted LSTM ELU | 0.9708 | 0.9794 | 0.9810 | 0.1026 | 0.0786 | 0.0841 |
| Boosted Cell State ReLU | 0.9667 | 0.9760 | 0.9805 | 0.1246 | 0.0887 | 0.0873 |
| Boosted Cell State ELU | 0.9632 | 0.9759 | 0.9783 | 0.1273 | 0.0958 | 0.0869 |

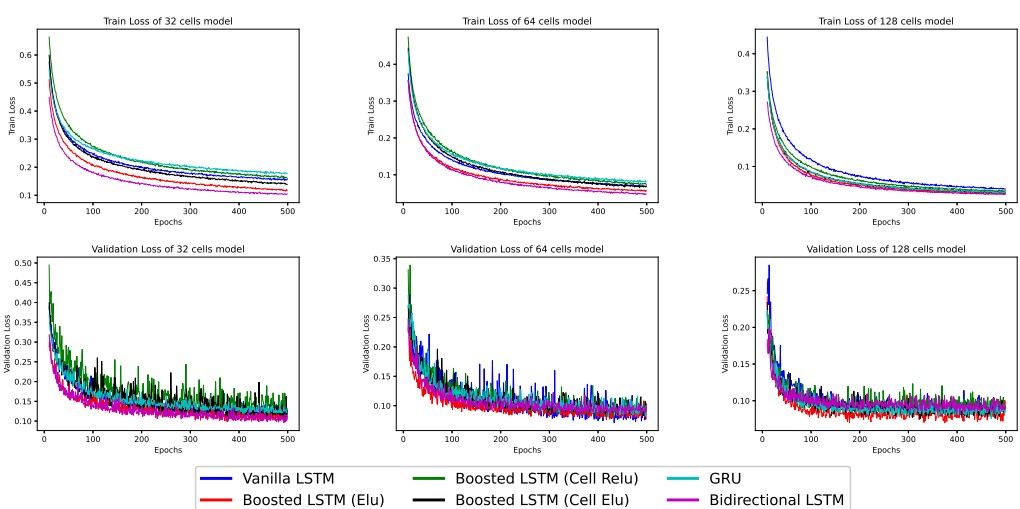

Figure 5: Loss vs. epochs for the train and validation datasets for all the considered permuted MNIST classification task architectures.

## 3.3 REUTERS DATASET

The training process on the Reuters dataset, shown in Fig. 7, consisted of 400 training epochs. The Boosted LSTM with the ELU-boosted activation function managed to constantly score the best results during the training process. It was also the fastest to converge for the 32-unit architecture, outperforming the Vanilla LSTM by 49,76%, to the point of overfitting, so it was necessary to apply an early stopping criterion to preserve its performance.

Table 3: The best accuracies and losses for final models of the Reuter's topics classification task.

| LSTM Architecture | Accuracy | | | Loss | | |
|---|---|---|---|---|---|---|
| | 8 Units | 16 Units | 32 Units | 8 Units | 16 Units | 32 Units |
| Vanilla LSTM | 0.6950 | 0.7039 | 0.7239 | 1.3237 | 1.3865 | 1.6182 |
| GRU | 0.7231 | 0.7012 | 0.7102 | 1.1817 | 1.3276 | 1.5412 |
| Bidirectional LSTM | 0.7346 | 0.7070 | 0.7320 | 1.1058 | 1.4588 | 1.6347 |
| Boosted LSTM ELU | 0.7462 | 0.7204 | 0.7516 | 1.1606 | 1.2544 | 1.5681 |
| Boosted Cell State ReLU | 0.7017 | 0.6670 | 0.7155 | 1.2255 | 1.4812 | 1.6875 |
| Boosted Cell State ELU | 0.7244 | 0.6768 | 0.7026 | 1.1843 | 1.5659 | 1.7077 |

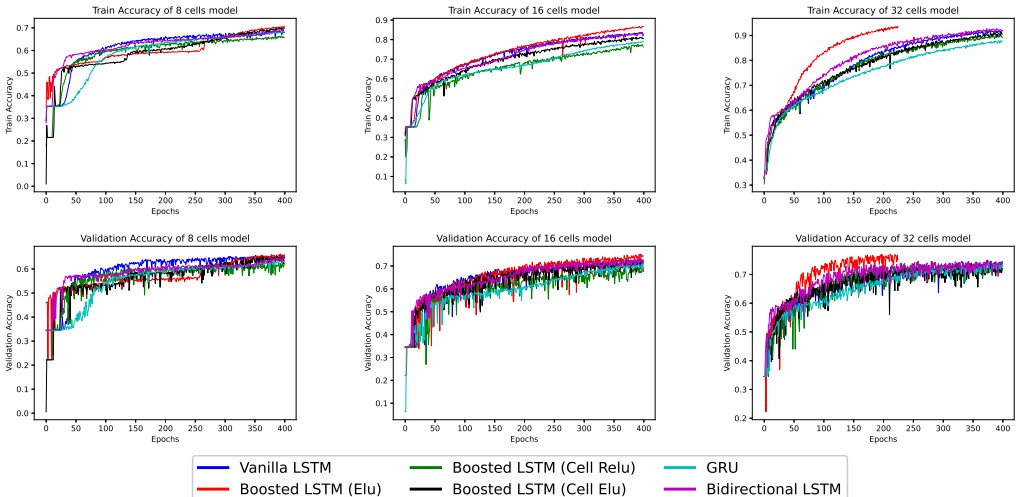

Figure 6: Accuracy vs. epochs for the train and validation datasets for all the considered Reuters topics classification task architectures. The difference in epochs is due to the response to the implemented early stopping criterion.

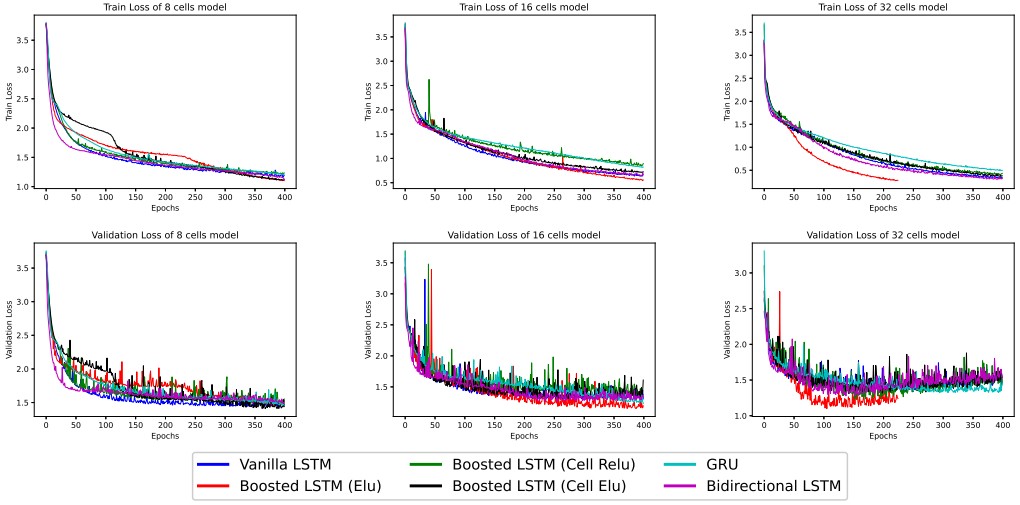

Figure 7: Loss vs. epochs (right) for the train and validation datasets for all the considered Reuters topics classification task architectures. The difference in epochs is due to the response to the implemented early stopping criterion.

After the final evaluation, which results are presented in Table 3, we found that the Boosted LSTM with ELU achieved an overall 8.81% better performance than Vanilla LSTM for the 8-units architecture, 2.34% better for the 16-units architecture, and 3,82% better for the 32-units architecture. When comparing holistically, the fully-boosted LSTM always achieved the best results in accuracy, and in most cases when considering loss. It is worth noting that models with only Cell State boosted performed tangibly worse in comparison to their competitors, being able to only beat the Vanilla LSTM and only for the 8-unit architecture.

## 4 DISCUSSION

The presented experiments have shown that the addition of extra layers with the ELU activation functions improved and accelerated the performance of the models tested on the three datasets mentioned.

We conducted research on many variations of differently boosted LSTM units, but the one that uses ELU as the boosted activation function for additional inside layers managed to consistently outperform the baseline LSTM and other compared architectures in nearly all of the experiments. This variation also exhibited a very fast convergence rate.

From all of the other architectures, only LSTMs with boosted Cell State using the ReLU and ELU function were able to perform close enough to the Vanilla LSTM, and only the ELU function was able to perform better in isolated cases.

The main limitation of the proposed boosted architectures is the increase in the computational complexity since it has more parameters and takes a longer time to train for the same number of epochs compared to a simple LSTM. Yet, due to their properties, their use can actually result in a fewer number of epochs overall required to obtain the same results, thus leading to faster training.

The new Boosted LSTM units introduced could form a good basis for more sophisticated models. This may lead to a boost in performance on more difficult tasks and datasets when using recurrent neural networks. Since the results achieved by the boosted architecture were already comparable to those of Bidirectional LSTM, adding solutions similar to the BiLSTM Forward and Backward States could be a promising direction in an attempt to further refine the Boosted LSTM. A great research opportunity could also lie in assessing the impact of similar changes applied to the Gated Recurrent Unit (Cho et al., 2014).

## 5 CONCLUSIONS

In this paper, we have introduced Boosted LSTM units, which enhanced Vanilla LSTM units by using additional inner layers to increase their performance. The presented experiments have shown that the Exponential Linear Units (ELU) used in these extra layers as activation functions allow the Boosted LSTM to outperform the baseline LSTM by even 7.37% while often reducing the number of epochs required to achieve similar results by 30.15–49.76%. This Boosted LSTM is also able to compete and succeed against other architectures, such as GRU and Bidirectional LSTM. We hope that our research could allow for the development of more efficient LSTM architectures, increasing model performance on many different tasks related to sequential data processing.

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
