# OpenReview forum: "Boosted Long Short-Term Memory with Additional Inner Layers"
_ICLR.cc/2024/Conference — Submitted to ICLR 2024_

### Official Review · Reviewer_ueko · 2023-10-26

**Soundness:** 1 poor
**Presentation:** 2 fair
**Contribution:** 1 poor
**Rating:** 1
**Confidence:** 5

**Summary:**

The authors propose a small modification to the standard LSTM model by adding the ELU activation function before the sigmoid and tanh activation functions.

Experiments are performed on three small datasets: IMDB sentiment classification, Permuted Sequential MNIST dataset, Reuters Newswire classification dataset.

The models in the experiments are tiny (hidden size of 2, 4 or 8 for one experiment).

Comparisons are done to the vanilla LSTM and GRU only.

**Strengths:**

Introduces a new LSTM variant which could potentially be better.

**Weaknesses:**

- The model sizes in the experiments are way too small to be relevant. Maybe you don't need larger models for these tasks, but then you should choose other tasks.
- More common tasks are not used. When introducing a new LSTM variant, the most canonical task would be language modeling, e.g. enwik8 or enwik9 or so.
- No comparison to current state-of-the-art results on the specific tasks. I'm quite sure (without checking) that the presented models are far behind.
- No comparison to Transformer models. Currently, the standard models are Transformer models, and when introducing any new model (even a new LSTM variant), you should compare to Transformer models as well. Why would I use this LSTM variant over the Transformer? The paper should tell me.
- It is being talked about computing time, but I don't actually see that being measured in the experiments. I would like to see a plot computing time vs accuracy. Number of epochs is not so useful, because as you point out, the vanilla LSTM is faster to train. (Note that the Transformer is even much faster to train, so that is why this comparison is also important here.)
- There is no code to reproduce the experiments?
- There is not much analysis.
- The motivation is weak. Why not e.g. add multiple LSTM layers, instead of adding layers inside the LSTM?

**Questions:**

From abstract:

> Numerous experiments have shown that adding more complexity to neural network architectures may lead to a significant increase in performance that outweighs the incurred costs of an upgraded structure.

Remove that. That is not really adding any information. Let the experiments just speak for themselves. Describe here in the abstract more on how "significant" the performance increase actually is.

Is this compared to Transformer or other recent SOTA models?

Abstract: "three empirical tasks" - it should say more about what tasks actually, and also give numbers on how much performance increases.

> Therefore, Boosted LSTM requires more resources to run, both memory and computational time- wise and its performance must justify the additional commitment.

So, is the computation time also measured? For given computation time, what model reaches what performance?

While the modification to the LSTM is described, which leads to the Boosted LSTM, the model itself is not described at all. I was initially thinking that this was used for language modeling, i.e. text as input via an embedding, then one or multiple Boosted LSTM layers, and then a softmax on top. Maybe with residual connections or so. But after looking at the experiments, this does not seem to be the case. But I don't find the actual models described in detail.

> The model consisted of an embedding layer, an appropriate LSTM layer, a dense layer of 8 units with a ReLU activation function, and a final dense layer with 1 unit and a sigmoid activation function.

Ok. Is there any reference why this type of model? What dimensions for the embedding?

> Tests were performed for models with 2, 4, and 8 hidden units.

Is this a joke?

If this task (I don't know this task) requires only such tiny models for good performance, then this task is really not relevant. You should choose some standard big benchmark, not a toy benchmark. E.g. take enwik8 or so for language modeling.

Standard realistic (and relevant) model sizes would be 512 to 2048 hidden dimensions, and having multiple layers, starting maybe with 3, up to 10, or maybe also going deeper up to 20 (if you have residual connections).

For MNIST, I don't see the model being described. How many layers?

> Hidden units of 32, 64, and 128 sizes were used during the experiments.

It's also quite small.

Reuters Newswire:

> hidden units (8, 16, 32)

Then this is also not really relevant.

Code published?

Compared to SOTA results?

**Details Of Ethics Concerns:**

-

---

### Official Review · Reviewer_xgqM · 2023-11-01

**Soundness:** 2 fair
**Presentation:** 3 good
**Contribution:** 2 fair
**Rating:** 3
**Confidence:** 2

**Summary:**

In this paper, the authors propose a Boosted LSTM model created by adding layers inside the LSTM unit to optimize the model by enhancing its memory and reasoning capabilities and evaluated the performance of different versions of Boosted LSTM architectures using three empirical tasks. The experiments have shown that the Boosted LSTM unit performs better than the similar models created from the simple LSTM units while often taking fewer epochs to achieve similar or better results.

**Strengths:**

The idea is simple and the paper is esay to follow.
The paper provided training details and also the training curves for the newly proposed models. And the author did ablation study to compare modifying different parts of the original LSTM model.

**Weaknesses:**

1) Baseline model and new models have different number of parameters and uses the same regulation parameters during training, so the comparison might not be fair.
2) The boosted LSTM model looks very similar to the original LSTM model, so I feel it's not a big improvement.
3) The tested datasets are small.

**Questions:**

Have you tried different regulation parameters can have a big impact on the model performance, and did you tune the parameter for the baseline models?

---

### Official Review · Reviewer_8bpo · 2023-11-01

**Soundness:** 1 poor
**Presentation:** 2 fair
**Contribution:** 2 fair
**Rating:** 3
**Confidence:** 4

**Summary:**

This paper introduces Boosted Long Short-Term Memory (Boosted LSTM) units. Compared to conventional LSTM units with forget gate, Boosted LSTM units contain an additional fully-connected hidden layer before the LSTM gate activations and the LSTM cell input. The paper evaluated different activation functions for the additional fully-connected hidden layer and the application of this hidden layer on all as well as individual LSTM gates and cell input. Three tasks, 2 word sequence and 1 image character recognition, were used for evaluation of the proposed units. These experiments showed a better performance of the Boosted LSTM, compared to the conventional LSTM units with forget gate, when employing the same number of LSTM units.

**Strengths:**

## Originality
1. The proposed Boosted LSTM unit was, to my knowledge, not published before.

## Quality
2. The paper uses 3 real-world datasets to assess the performance of the Boosted LSTM.
3. Both final results and training behavior are reported in the paper.
4. Different versions of Boosted LSTM (hidden layer before all as well as individual LSTM inlets) are investigated.
5. A comparison of complexity between vanilla LSTM and Boosted LSTM is performed based on O-notation.

## Clarity
6. The proposed idea in the paper is easy to follow.

## Significance
7. If the Boosted LSTM can provide substantial performance improvements, this paper would be of high relevance to the RNN community. This would make it significant. However, at this point the extent of the experimental analysis, the setup, and the results are not sufficiently convincing in my opinion.

**Weaknesses:**

## Originality
8. In the 1997 LSTM paper [1], Hochreiter et al already mention the potential of addition of hidden units and more flexible changes to the LSTM blocks (paragraphs *Network topology*, *Memory cell blocks*). Additionally, as also pointed out by the authors, various modifications to LSTM units were made since their introduction. The addition of hidden layers in the LSTM block is, from my point of view, not a very original concept. The paper would gain in originality if a strong empirical analysis was performed, which in my opinion, is not yet the case.

## Quality
9. A major weakness in the comparison of LSTM architectures in the paper is that there is no fair comparison w.r.t. trainable parameters or model complexity. One can observe that the model performance increases with the number of used units for vanilla and Boosted LSTM in the reported tables. This indicates that model complexity is a major driver of predictive performance. To allow for a fair comparison, the experimental setup would, in my opinion, need to contain:
    1. Vanilla LSTM versions with increased numbers of LSTM units, such that the trainable parameters are close to the Boosted LSTM version.
    2. A comparison to a model that contains a fully-connected hidden layer before the Vanilla LSTM. This would be equivalent to having shared weights in the elu units in Fig1b.
    3. Same as the previous point but with a fully-connected hidden layer after the Vanilla LSTM.
10. The reported results are not very convincing in their current form. Relying on only three datasets in an empirically motivated publication would only be convincing to me if the results were significantly better for Boosted LSTM compared to Vanilla LSTM. I would strongly suggest to employ a statistical test to report whether the performance improvement is significant.
11. In my opinion, the paper would strongly profit from a broader investigations of modifications and hyperparameters:
    1. I understand that the paper investigates modifications by introducing fully-connected hidden layers. However, this raises the question of the impact of multiple stacked hidden layers. Especially for the SELU activation function, which could, according to it’s theory, profit from larger and deeper networks.
    2. If I understand correctly, only the RMSprop optimizer was used. I would strongly suggest to investigate SGD with momentum and AdamW, as the choice of optimizer might impact the training of more complex architectures like this.

## Clarity
12. The authors state that “This allowed us to test different capabilities of the architecture while maintaining focus on the text inference tasks.”.  I found this statement confusing. The focus on text inference tasks was not stated or motivated as focus before, why bring this up here? Furthermore, Permuted MNIST is not text inference but a vision task.
13. The “embedding layer” used is not described in detail. A figure depicting the over-all architecture would also be helpful.
14. How is early stopping performed here?
15. Fig4 would be very interesting but unfortunately is hardly readable due to over-plotting. I would suggest to rework this figure.
16. “Other hyperparameters were adapted accordingly to the dataset considered, taking the Vanilla LSTM performance as a reference point for their refinement.” - If I understand correctly, this results in a potential underestimation of the Boosted LSTM performance. If so, I would suggest to explicitly state this.
17. “Both after the embedding layer and after the LSTM layer, 0.25 dropouts were used,” - which type of dropout? Are the remaining values rescaled?
18. Does the l2 regularization also affect the elu layers?
19. Are the test sets balanced to warrant a comparison via Accuracy and loss?
20. “Tests were performed for models with 2, 4, and 8 hidden units.” - At this point in the paper, it is unclear whether “hidden units” refers to LSTM units or other units.
21. Since “Boosting” already has different connotations in ML, I would suggest to change the method name. This is just my personal opinion, tough.

## Significance
22. If the Boosted LSTM can provide substantial performance improvements, this paper would be of high relevance to the RNN community and would make this a significant contribution. However, at this point the experimental analysis and results are not sufficiently convincing in my opinion.

## References

[1] Sepp Hochreiter and Jürgen Schmidhuber. Long Short-Term Memory. Neural Computation, 9 (8):1735–1780, 11 1997. ISSN 0899-7667. doi: 10.1162/neco.1997.9.8.1735. URL https://doi.org/10.1162/neco.1997.9.8.1735.

[2] Klaus Greff, Rupesh K. Srivastava, Jan Koutnik, Bas R. Steunebrink, and Jurgen Schmidhuber. LSTM: A search space odyssey. IEEE Transactions on Neural Networks and Learning Systems, 28(10):2222–2232, oct 2017. doi: 10.1109/tnnls.2016.2582924. URL https://doi.org/10.1109%2Ftnnls.2016.2582924.

**Questions:**

The main limitations of the paper in my opinion are points 9. and 10. above. Other questions to the authors are listed in the previous sections.

---

### Meta-Review · Area_Chair_Ua62 · 2023-12-08

**Metareview:**

The paper presents an approach to enhance LSTM units coined Boosted LSTM. This involves integrating additional layers within the LSTM unit. The authors experimented with various configurations of these Boosted LSTM units, focusing on the placement of additional layers, the choice of activation functions in these layers, and the size of the model's hidden units. The paper also explores the impact of applying the additional fully-connected hidden layer to different components of the LSTM unit.

**Justification For Why Not Higher Score:**

It is a method paper but, as pointed out by all reviewers, the empirical experiments are clearly not up to the standard of ICLR -- they are simply not convincing. The authors have chosen to not engage in discussions with the reviewers. The reviewers unanimously and strongly proposed a rejection.

**Justification For Why Not Lower Score:**

NA

---

### Decision · Program_Chairs · 2024-01-16

Reject